**Data Availability Statement:** All relevant data are within the manuscript and its Supporting Information files.

**Funding:** This research was funded by the project "RAISE - Robotics and AI for Socio-

# EMG-assessed paratonia: A novel approach to investigating motor response inhibition in healthy subjects

**Luca Puce**[1], **Lucio Marinelli**[1,2]*, **Antonio Currà**[3], **Laura Mori**[1,2], **Cristina Schenone**[1], **Filippo Cotellessa**[1], **Antonella Tatarelli**[4], **Daniele Pucci**[4], **Nicola Luigi Bragazzi**[5], **Carlo Trompetto**[1,2]

**1** Department of Neuroscience, Rehabilitation, Ophthalmology, Genetics, Maternal and Child Health (DINOGMI), University of Genoa, Italy, **2** IRCCS Ospedale Policlinico San Martino, Genoa, Italy, **3** Academic Neurology Unit, A. Fiorini Hospital, Terracina, LT, Department of Medico-Surgical Sciences and Biotechnologies, "Sapienza" University of Rome, Italy, **4** Artificial and Mechanical Intelligence Research Line, Istituto Italiano di Tecnologia (IIT), Center for Robotics and Intelligent Systems, Genova, Italy, **5** Department of Mathematics and Statistics, Laboratory for Industrial and Applied Mathematics (LIAM), York University, Toronto, Canada

* lucio.marinelli@unige.it

## Abstract

Paratonia is an involuntary muscle activity that occurs during passive joint mobilization and is common in people with dementia. It includes oppositional paratonia, in which muscle activity resists passive movement, and facilitatory paratonia, in which it assists movement. This phenomenon reflects a defect in motor response inhibition. In a recently published paper, we demonstrated that paratonia can be detected using surface electromyography (EMG) not only in patients with dementia but also in healthy individuals, the majority of whom do not exhibit clinically observable paratonia. This finding suggests that *EMG-assessed paratonia* may provide a novel approach to studying motor response inhibition in healthy subjects. The present study investigates this possibility for the first time. We recruited 120 healthy subjects under the age of 30, divided equally into three groups: sedentary, amateur, and professional athletes with low, moderate, and high levels of physical activity, respectively. Paratonia was assessed in the triceps and biceps brachii muscles during passive forearm movements performed manually. The results indicate that paratonia is more pronounced during fast and continuous passive movements, with facilitatory paratonia being more prevalent than oppositional paratonia. It is also more pronounced in the biceps than in the triceps. These findings, which mirror those previously observed in patients with dementia, suggest a similarity between paratonia in healthy subjects and those with cognitive impairment, supporting the hypothesis that paratonia in healthy individuals represents a form of impaired motor response inhibition. Furthermore, the comparison between groups showed that paratonia decreased with increasing physical activity, being least evident in athletes, more noticeable in amateurs, and most pronounced in sedentary individuals. This pattern confirms a key feature of motor response inhibition that has been shown in studies using traditional methods. Overall, our findings suggest that *EMG-assessed paratonia* provides a new method for studying motor response inhibition in healthy individuals.

economic Empowerment" and "Fit4MedRob - Fit for Medical Robotics". The funders had no role in study design, data collection and analysis, decision to publish, or preparation of the manuscript.

# 1. Introduction

Paratonia is the inability to relax muscles during assessment of muscle tone in the absence of spasticity and parkinsonian rigidity [1, 2]. It is traditionally associated with dementia [3–6], although mild forms of paratonia can be found in cognitively unimpaired individuals, particularly in the elderly [6–8].

Two distinct forms of paratonia can be observed: oppositional paratonia, where the subject resists passive movements despite being asked to remain relaxed, and facilitatory paratonia, where the subject involuntarily assists these movements, moving in the same direction as the examiner [9]. Both forms of paratonia are characterized by an impaired ability to reach and maintain muscle relaxation, inhibiting unwanted motor activity during muscle tone assessment. It is largely accepted that this impairment represents a manifestation of defective motor response inhibition due to frontal lobe dysfunction [9–13].

Motor response inhibition is a pivotal aspect of inhibitory control, used to suppress inappropriate behavior and interrupt unnecessary or irrelevant actions [14]. This process, controlled by frontal cortical circuits, is fundamental to ensuring effective adaptation to changing contexts, a skill that is critical in the world of sport [15].

Go/no-go and stop-signal tasks are paradigms specifically designed to examine motor response inhibition [14, 16, 17]. Research using these paradigms has shown that physical activity can improve motor response inhibition [18]. This effect may be due to the way in which physical activity, particularly in dynamic and unpredictable environments, enhances executive function and motor control, both of which are critical for motor response inhibition. Consistent with this, open-skill sports, which require adaptation to unpredictable contexts, have been found to promote motor response inhibition more effectively than static, closed-skill sports [19].

In 2017, we developed and validated a quantitative approach using surface electromyography (EMG) to assess both oppositional and facilitatory forms of paratonia in the elbow flexor and extensor muscles of individuals with dementia [20]. Paratonia measured by this method is hereinafter referred to as *EMG-assessed paratonia*. We showed that *EMG-assessed paratonia* in individuals with dementia is more pronounced during fast than slow passive movements, is more pronounced during continuous than discontinuous passive movements, is more prevalent in the Biceps Brachii (BB) than in the Triceps Brachii (TB), and the facilitatory activity is greater than the oppositional one [7, 20].

Moreover, we found that *EMG-assessed paratonia* is present in the large majority of healthy subjects, including young people. In these subjects, *EMG-assessed paratonia* is usually too small in amplitude to be clinically detectable, but otherwise has the same characteristics as *EMG-assessed paratonia* detected in individuals with dementia [7]. These findings show that the efficiency of frontal cortical circuits aimed to prevent involuntary muscle activation during tone assessment is far from being perfect not only in patients with cognitive impairment but also in healthy subjects, including young people, suggesting that *EMG-assessed paratonia* could be used to test motor response inhibition also in cognitively unimpaired individuals [7].

The present study was designed to validate this hypothesis. In order to do so, firstly, we investigated the neurophysiological characteristics *of EMG-assessed paratonia* in cognitively unimpaired healthy subjects to verify their correspondence with those observed in patients with dementia. Secondly, we tested whether the investigation of *EMG-assessed paratonia* in healthy subjects is able to reveal a well-known feature of motor response inhibition, specifically its correlation with the level of physical activity.

## 2. Materials and methods

### 2.1. Study design and ethics approval

This is a cross-sectional case-control study. The examinations and analyses of all subjects were performed at the University Hospital "IRCCS Ospedale Policlinico San Martino", Genoa, Italy.

The study protocol was reviewed and approved by the local ethics committee of the University of Genoa, Genoa, Italy (protocol number 2024.36, dated April 12, 2024) and registered on the public website ClinicalTrials.gov (identifier: NCT06573918). In addition, all participants provided written informed consent prior to their participation in the study. Recruitment of participants began on April 18, 2024, and ended on July 23, 2024.

### 2.2. Participants

Participants were eligible for the study if they were healthy and under 30 years of age. Exclusion criteria included the presence of pain in the flexor or extensor muscles of the arm; the use of muscle stimulants, relaxants, or steroids; and the use of tobacco, alcohol, or other substances.

The study grouped participants on the basis of their self-reported weekly physical activity level, measured in MET (Metabolic Equivalent of Task) according to Pierce's guidelines [21]. To ensure a balanced distribution, stratification was applied to the three groups without the use of additional methods such as blocking or minimization.

A total of 120 subjects were recruited and equally distributed among the three groups as follows: low active (17 men and 23 women, age 23.78 ± 3.87 years, BMI 25.34 ± 5.16 kg/m$^2$, MET 11.58 ± 3.05 hours/week), moderately active (29 men and 11 women, age 25. 33 ± 4.08 years, BMI 22.94 ± 3.48 kg/m$^2$, MET 21.33 ± 3.30 hours/week) and highly active (29 men and 11 women, age 24.13 ± 5.41 years, BMI 22.12 ± 2.33 kg/m$^2$, MET 117.00 ± 19.78 hours/week).

Within the highly active group, the participants were further subdivided according to the type of sport they practiced. Specifically, 20 participants were involved in open-skill sports such as tennis (five men and five women, age 25.30 ± 4.74 years, BMI 21.61 ± 2.57 kg/m$^2$, MET 107.50 ± 9.47 hours/week) and basketball (seven men and three women, age 23.50 ± 4.14 years, BMI 23.02 ± 2.48 kg/m$^2$, MET 103.00 ± 7.56 hours/week). The other 20 participants were involved in closed-skill sports such as swimming (four men and six women, age 24.20 ± 4.64 years, BMI 21.28 ± 1.60 kg/m$^2$, MET 147.20 ± 9.70 hours/week) and running (seven men and three women, age 23.50 ± 7.96 years, BMI 22.59 ± 2.45 kg/m$^2$, MET 110.30 ± 8.60 hours/week).

The low active participants were recruited from sedentary individuals working in corporate offices, call centers, or academic environments. The moderately active participants were recruited through amateur sports clubs, while the highly active participants were selected from professional sports teams. Depending on the specific context in which they were recruited, participants are referred to in this study as "sedentary," "amateur," and "professional athletes," respectively.

To accurately estimate participants' weekly MET hours, a detailed questionnaire was administered to collect comprehensive information on the physical activities performed during a typical week, including the duration and intensity of each activity. The metabolic equivalent for each activity was calculated using the Compendium of Physical Activities for Adults 2024 [22].

### 2.3. Sample size calculation

An *a priori* sample size and power analysis was carried out using the open-source G*Power (version 3.1.9.7). Assuming an alpha error probability of 0.05 and 80% power, with three groups and an effect size f of 0.40, the computation yielded an overall sample size of 66 [23].

## 2.4. EMG-assessed paratonia

Participants were informed about the type of experimental protocol to which they would be subjected. However, the experimenters administering the interventions and those evaluating the outcomes were blinded to the assignment of conditions. Blinding was ensured by a third person who managed the assignments randomly.

To assess *EMG-assessed paratonia*, we recorded sEMG signals using a bipolar wireless system (Cometa Srl, Milan, Italy) with a sample frequency of 2000 Hz placed on the BB and TB muscles of the dominant side of the participant's body, following the recommendations of the SENIAM (Surface Electromyography for Noninvasive Assessment of Muscles) protocol for electrode placement [24]. An electronic goniometer (model TSD130B, Biopac Systems Inc, USA) placed directly on the elbow joint was used to continuously monitor the elbow joint angle.

The experiment implemented a previously outlined experimental paradigm [7, 20]. In a quiet room with a controlled temperature of 22°C and humidity of ≈50%, the examinee was subjected to four blocks of 15 passive elbow flexion-extension movements performed by the physiatrist (F.C.) in random order, comfortably seated in a chair with eyes closed and muscles relaxed. The movements were initiated from a fully extended position at 0 degrees of extension to a maximum flexion of 150 degrees [25]. These series included 1) 15 continuous movements at 40 beats per minute (BPM); 2) 15 continuous movements at 100 BPM; 3) 15 discontinuous movements at 40 BPM; 4) 15 discontinuous movements at 100 BPM. The 40 BPM cadence was selected to simulate slow and precise movements characteristic of certain sports activities. Conversely, the 100 BPM cadence was chosen to represent rapid and explosive movements typical of other sports contexts. We deliberately avoided using higher movement speeds to prevent the elicitation of phasic reflexes, which could interfere with and obscure the assessment of paratonic activity.

The cadence of the movements was adjusted so that the points of maximum elbow flexion and maximum elbow extension corresponded to two consecutive beats of the metronome, set according to the specific velocity of each block.

To introduce discontinuous movements, the experimenter waited for a random interval of metronome beats (1 to 4) while holding a position of maximum flexion or extension before proceeding to the next movement (Fig 1A).

*EMG-assessed paratonia* recorded during flexion in the BB was associated with the facilitatory form of paratonia, whereas recordings during extension were associated with the oppositional form. Conversely, in the TB, *EMG-assessed paratonia* recorded during flexion indicated the oppositional form, while recordings during extension indicated the facilitatory form (Fig 1B).

Between each block, the examiner provided verbal encouragement to maintain the participant's maximum relaxation.

## 2.5. EMG processing

The raw EMG signal was first band-pass filtered in a specific frequency range (20–450 Hz). They were then rectified and smoothed using a 4th order Butterworth low pass filter with a cut-off frequency of 5 Hz [26].

Each EMG signal corresponding to a movement of a given block was normalized with respect to the maximum voluntary isometric contraction of the muscle studied. The data were then interpolated to 101 points for each flexion and extension phase and averaged [27].

Finally, to obtain a complete measure of muscle activity that considers both the amplitude and duration of the involuntary contractions of the muscle, we calculated the area under the EMG signal curve using the trapezoidal method [28]. This was done for the computation of the area under the signal curve during the flexion and extension periods.

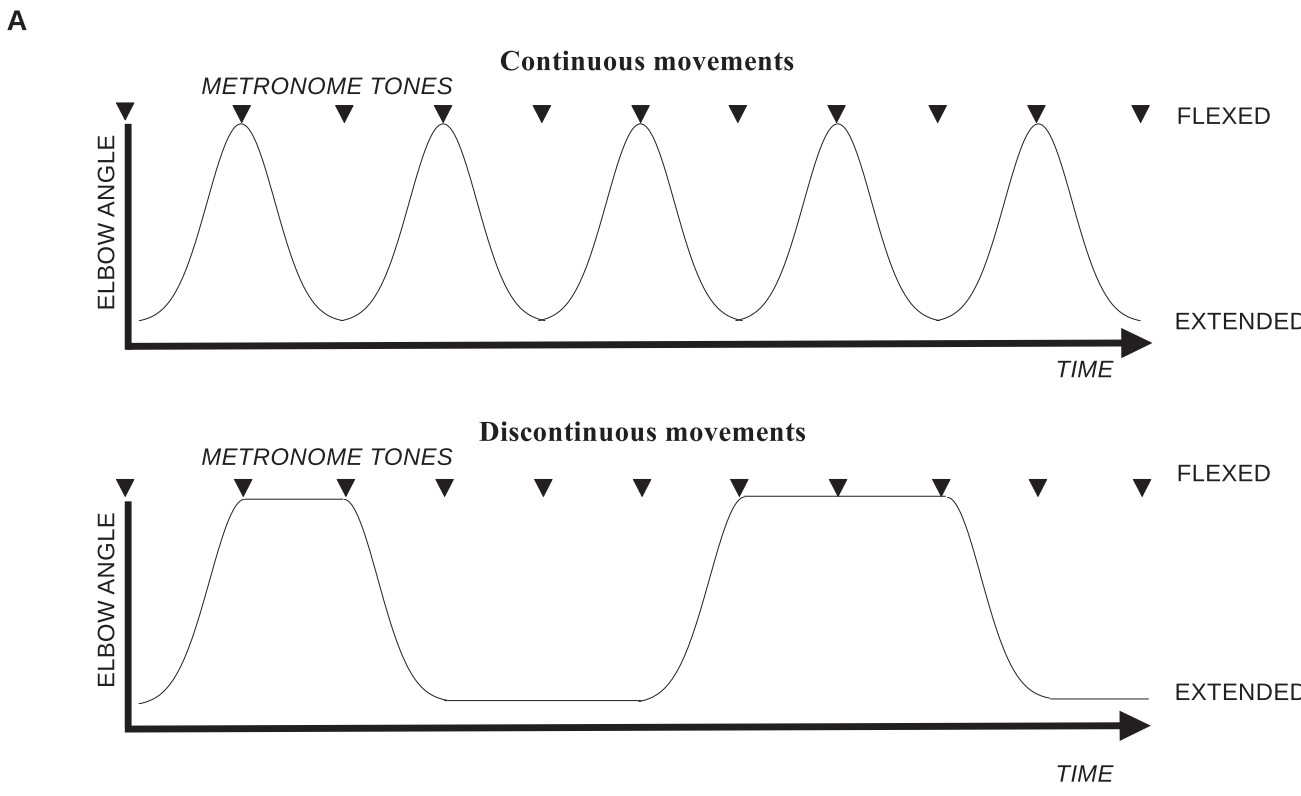

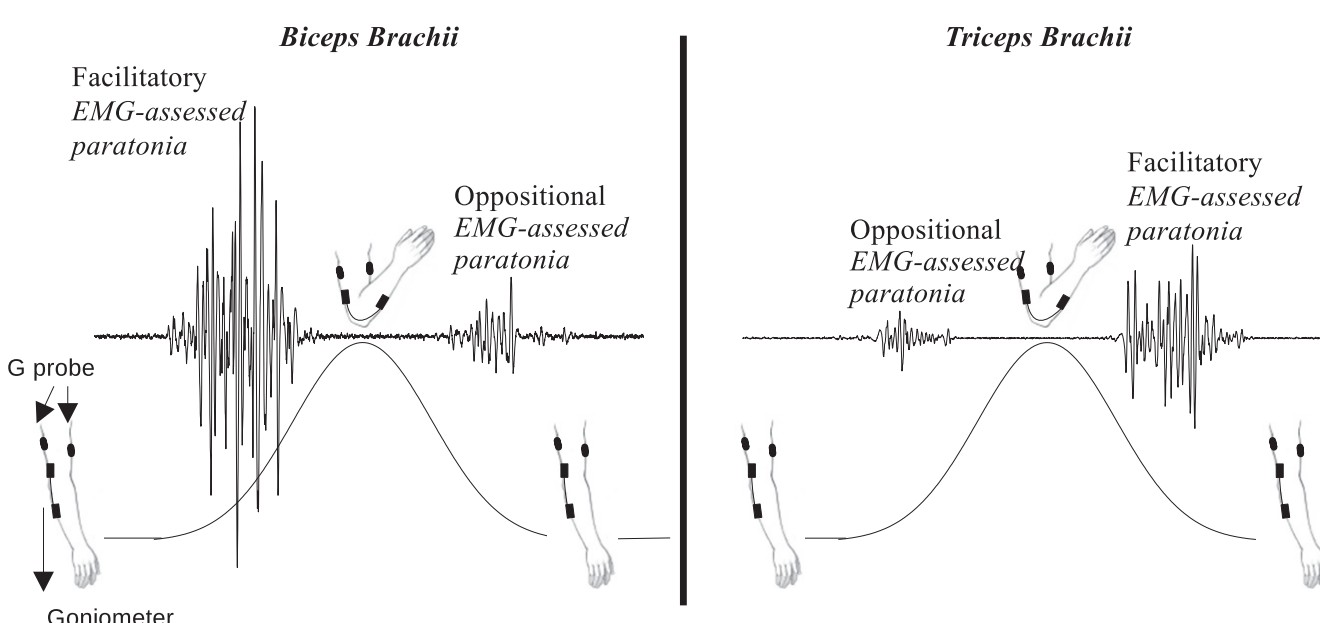

**Fig 1.** (A) Depicts passive elbow flexion and extension movements performed either in a continuous or discontinuous manner, synchronized with metronome beeps. (B) Shows the detection of the facilitatory form of EMG-assessed paratonia in the biceps brachii during passive flexion and in the triceps brachii during passive extension. Conversely, the oppositional form of EMG-assessed paratonia is observed in the biceps brachii during passive extension and in the triceps brachii during passive flexion.

## 2.6. Statistical analysis

The Kolmogorov–Smirnov and Shapiro–Wilk tests were used to verify the normal distribution of the data. The unpaired two-sample t test (t-test) or Mann-Whitney (MW) test were used to verify the presence of significant differences between TB and BB muscles area in four conditions (40 BPM vs. 100 BPM, Discontinuous vs. Consecutive, BB vs. TB and Facilitatory vs. Oppositional). The statistical significance was established for $p < 0.05$.

A one-way ANOVA was performed to evaluate the effect of the fitness level differences and the differences between disciplines. When relevant differences were observed in the ANOVA, we performed post-hoc analyses with Bonferroni's corrections. All analyses were performed using Matlab (MathWorks, Natick, MA, USA).

# 3. Results

The study did not experience any protocol deviations as all participants completed the intended intervention and adhered to the treatment protocol and were included in the final analysis.

The eligibility, enrollment, allocation, and analysis processes are pictorially shown in Fig 2.

The three groups exhibited highly significant differences in physical activity levels (Fisher's F = 984.46, $p < 0.001$). Post hoc analysis indicated that professional athletes had significantly higher MET values compared to both sedentary and amateur participants (all pairwise comparisons, $p < 0.001$). Furthermore, amateur athletes demonstrated higher MET scores than sedentary individuals ($p < 0.001$). Among the four groups of professional athletes, significant differences in physical activity levels were also observed (F = 45.15, $p < 0.001$). Post hoc analysis revealed that swimmers had significantly higher MET values compared to athletes in other sports (basketball, tennis, and running), with each comparison yielding a p value of 0.001.

## 3.1. Neurophysiological characteristics of EMG-assessed paratonia

The *EMG-assessed paratonia* was lower during slow passive movements (40 BPM) than during fast passive movements (100 BPM), both when the muscle was subjected to shortening (facilitatory paratonia) and when it was subjected to lengthening (oppositional paratonia). Specifically, for facilitatory paratonia, the TB showed a mean difference (MD) of -0.68% ($p < 0.001$, effect size [ES] = -0.57), while the BB showed a MD of -0.72% (p = 0.009, ES = -0.34). For oppositional form, the TB showed a MD of -0.13% (p = 0.01, ES = -0.22), and the BB showed a MD of -0.18% (p = 0.02, ES = -0.29) (Fig 3).

Discontinuous passive movements compared to continuous passive movements had lower *EMG-assessed paratonia* in both facilitatory form (TB: MD = -0.41%, p = 0.02, ES = -0.35; BB: MD = -0.68%, p = 0.02, ES = -0.32) and oppositional form, the latter limited to BB (MD = -0.34%, p = 0.002, ES = -0.53) (Fig 4).

Facilitatory form outperformed oppositional form with an MD of 1.63% (p<0.001) and an ES of 1.55.

The *EMG-assessed paratonia* of BB was higher than of TB, with an MD of 1.4% (p<0.001, ES = 0.89) for facilitatory form and 0.22% (p<0.001, ES = -0.43) for oppositional form (Fig 5).

## 3.2. EMG-assessed paratonia in professional athletes, amateurs and sedentary people

A significant effect was found in the interactions between the three groups studied for *EMG-assessed paratonia* in BB and TB (Fig 5). Specifically, for the facilitatory form, BB showed an effect with F = 136.00, $p < 0.001$ and TB showed an effect with F = 112.07, $p < 0.001$. For the

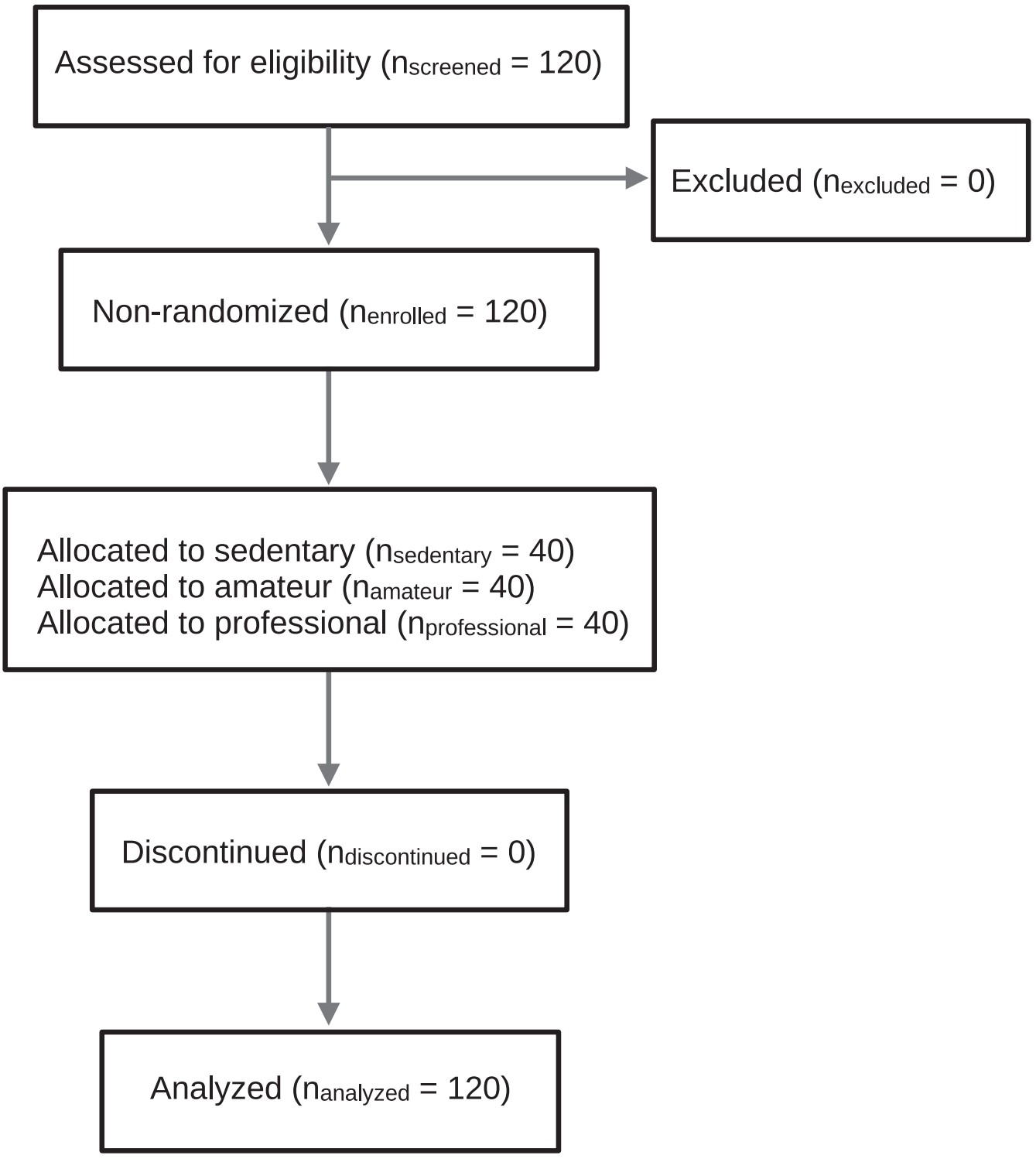

**Fig 2. TREND flow diagram.** Description of study population recruitment.

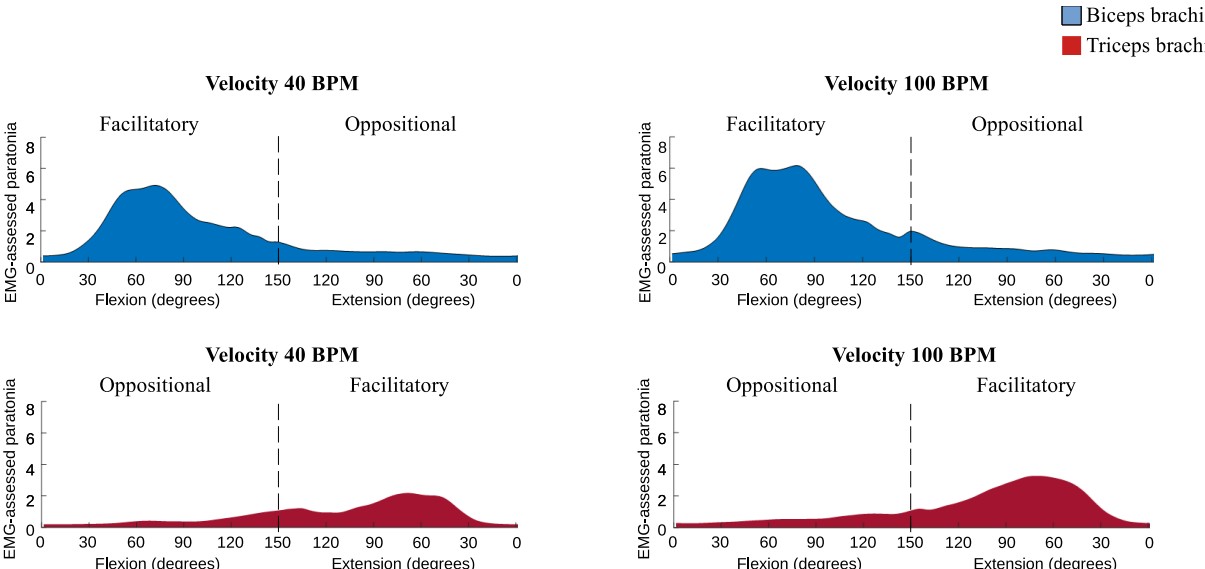

**Fig 3. Facilitatory and oppositional EMG-assessed paratonia in the biceps brachii (blue) and triceps brachii (red) during passive elbow flexion and extension at different metronome velocities (40 BPM vs. 100 BPM).**

oppositional form, BB showed an effect with F = 51.80, p < 0.001 and TB showed an effect with F = 53.31, p < 0.001 (Fig 6).

Post-hoc analysis of facilitatory form in BB revealed the following MD: between sedentary and amateurs, the MD was 2.06% (p < 0.001, ES = 1.67); between sedentary and professional athletes, the MD was 3.89% (p < 0.001, ES = 3.07); and between professional athletes and amateurs, the MD was -1.82% (p < 0.001, ES = -3.43). Similarly, examining facilitatory form in TB, the MD between sedentary and amateurs was 1.19% (p < 0.001, ES = 1.61), and between

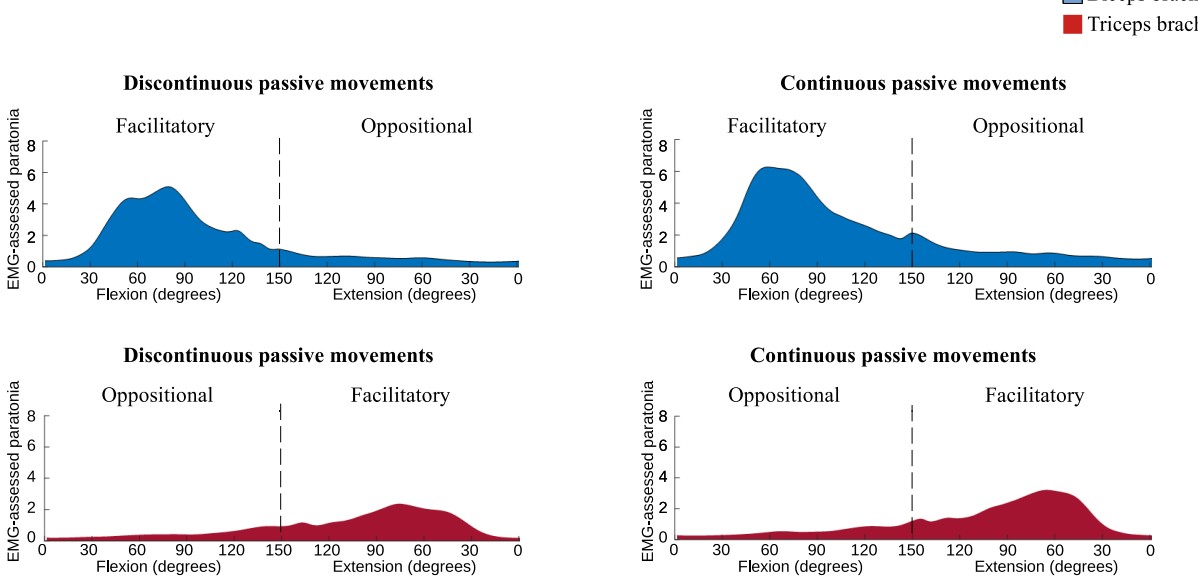

**Fig 4. Facilitatory and oppositional EMG-assessed paratonia in the biceps brachii (blue) and triceps brachii (red) muscles during passive elbow flexion and extension, comparing discontinuous and continuous passive movements.**

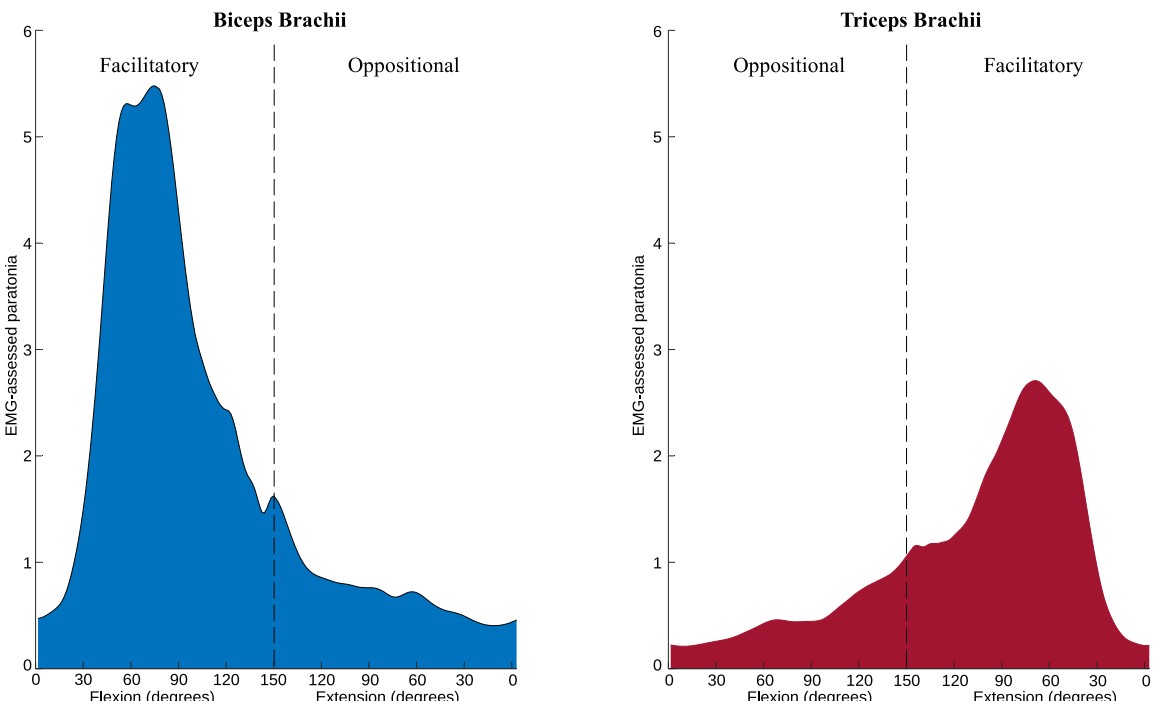

**Fig 5.** Facilitatory and oppositional EMG-assessed paratonia in the biceps brachii (left) and triceps brachii (right) muscles during passive elbow flexion and extension.

sedentary and professional athletes, the MD was 2.17% (p < 0.001, ES = 3.36). The comparison between professional athletes and amateurs showed an MD of -0.98% (p < 0.001, ES = -1.7).

For the oppositional responses of BB, an MD of 0.5% p < 0.001, ES = 1.11) was observed between sedentary individuals and amateurs, and 0.87% (p < 0.001, ES = 2.03) between sedentary individuals and athletes. The comparison between athletes and amateurs showed an MD of -0.36% (p < 0.001, ES = -1.46). In the case of the oppositional responses in TB, the MD between sedentary individuals and amateurs was 0.66% (p < 0.001, ES = 1.44), and between sedentary individuals and athletes, it was 0.82% (p < 0.001, ES = 1.9). No statistically significant differences were found between athletes and amateurs.

### 3.3. EMG-assessed paratonia in open and closed skill spots

*EMG-assessed paratonia* was higher in closed-skill sports compared to open-skill sports. Specifically, for the BB, the MD was 0.78% with an ES of 1.68, and for the TB, the MD was 0.69% with an ES of 2.34, both showing significant differences (p<0.001) in the facilitatory form. A similar significant trend was observed for the oppositional form: the BB had an MD of 0.16% (p = 0.01, ES = 1.12), while the TB had an MD of 0.15% (p<0.001, ES = 1.57). A sport-specific analysis is provided in Table 1 and Fig 7.

## 4. Discussion

### 4.1. Neurophysiological characteristics of EMG-assessed paratonia in cognitively unimpaired individuals (healthy subjects)

*EMG-assessed paratonia* was found to be greater during fast passive movements (100 BPM) compared to slow passive movements (40 BPM) and during continuous passive movements

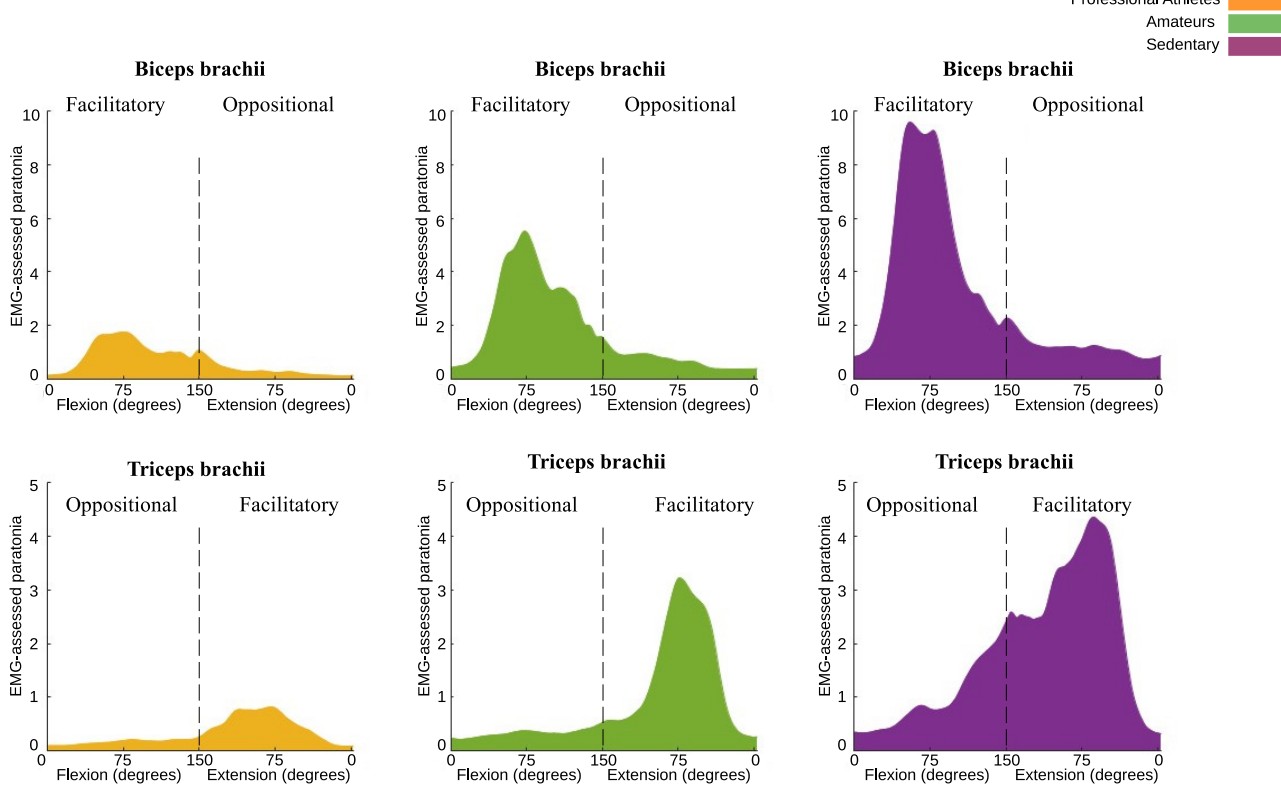

**Fig 6.** Facilitatory and oppositional EMG-assessed paratonia in the biceps brachii (top row) and triceps brachii (bottom row) muscles during passive elbow flexion and extension across three groups: professional athletes, amateurs, and sedentary individuals.

compared to discontinuous ones. Additionally, *EMG-assessed paratonia* was more pronounced in muscles undergoing passive shortening (facilitatory paratonia) compared to those undergoing passive lengthening (oppositional paratonia). Finally, *EMG-assessed paratonia* was more significant in the BB muscle compared to the TB muscle.

These characteristics of *EMG-assessed paratonia* in healthy subjects confirm in a much larger population our previous results and show that the EMG activity recorded during muscle tone assessment in healthy subjects has the same characteristics as the EMG activity observed

**Table 1. Comparison of facilitatory and oppositional EMG-assessed paratonia in the biceps brachii and triceps brachii muscles across different sports: Basketball, tennis, running, and swimming.**

| EMG-assessed paratonia | Basketball *vs.* Tennis | | | Basketball *vs.* Running | | | Basketball *vs.* Swimming | | | Tennis *vs.* Running | | | Tennis *vs.* Swimming | | | Running *vs.* Swimming | | |
|---|---|---|---|---|---|---|---|---|---|---|---|---|---|---|---|---|---|---|
| | MD % | *p* value | ES | MD % | *p* value | ES | MD % | *p* value | ES | MD % | *p* value | ES | MD % | *p* value | ES | MD % | *p* value | ES |
| *Biceps Brachii* | | | | | | | | | | | | | | | | | | |
| **Facilitatory** | -0.52 | 0.004 | -2.14 | -1.44 | <0.001 | -4.05 | -0.65 | <0.001 | -2.95 | -0.92 | <0.001 | -2.3 | -0.13 | 0.78 | -0.44 | 0.78 | <0.001 | 1.99 |
| **Oppositional** | -0.08 | 0.7 | -0.73 | -0.33 | <0.001 | -1.63 | -0.06 | 0.78 | -0.79 | -0.26 | 0.004 | -1.19 | 0.01 | 0.9 | 0.09 | 0.27 | 0.003 | 1.29 |
| *Triceps Brachii* | | | | | | | | | | | | | | | | | | |
| **Facilitatory** | -0.19 | 0.09 | -2.66 | -1.1 | <0.001 | -4.65 | -0.47 | <0.001 | -2.2 | -0.91 | <0.001 | -3.7 | -0.29 | 0.003 | -3.03 | 0.62 | <0.001 | 2.55 |
| **Oppositional** | -0.08 | 0.08 | -2.2 | -0.27 | <0.001 | -2.68 | -0.12 | 0.005 | -1.32 | -0.18 | <0.001 | -1.73 | -0.04 | 0.65 | -0.75 | 0.14 | <0.001 | 1.36 |

Abbreviations: MD = mean difference; ES = effect size.

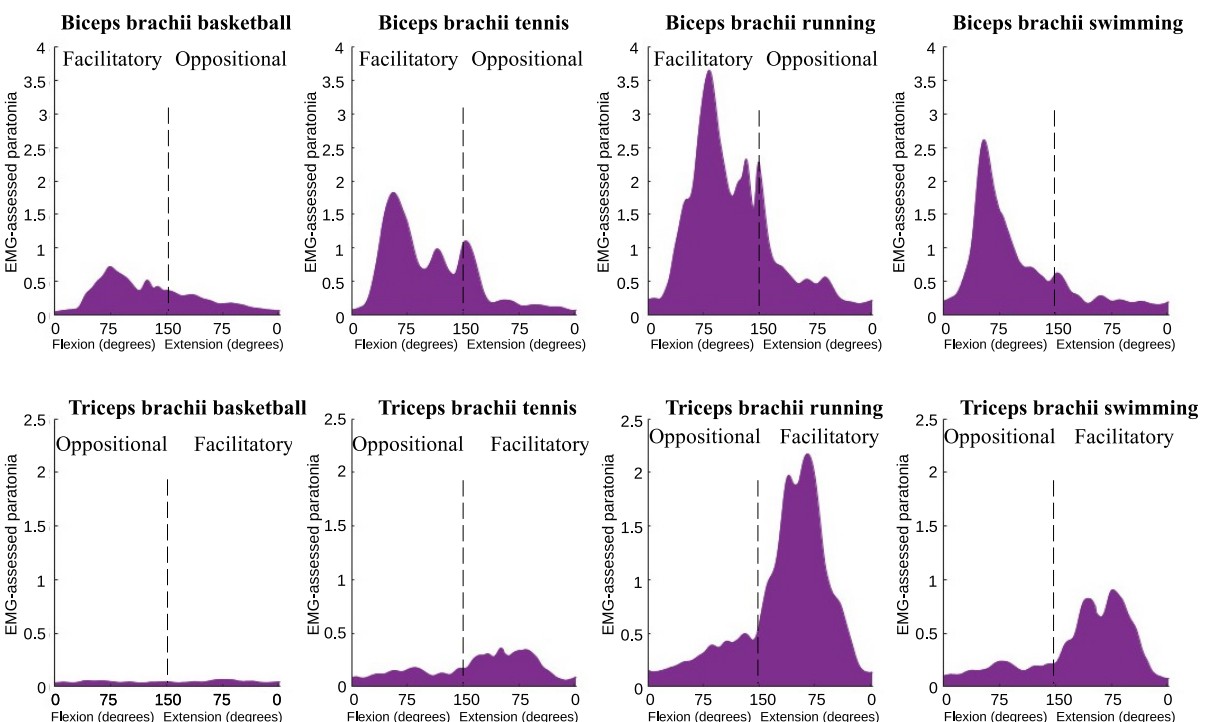

**Fig 7.** Facilitatory and oppositional EMG-assessed paratonia in the biceps brachii (top row) and triceps brachii (bottom row) muscles during passive elbow flexion and extension across different sports: basketball, tennis, running, and swimming.

in paratonic subjects with dementia [7, 20]. This means that *EMG-assessed paratonia* in healthy subjects can be equated with paratonia observed in patients with dementia, with the only difference that the amplitude of EMG activity in healthy subjects is generally insufficient to be identified as paratonia during a clinical examination of muscle tone.

Therefore, like paratonia in patients with dementia, *EMG-assessed paratonia* in healthy subjects can be viewed as a form of defective motor response inhibition. The milder the *EMG-assessed paratonia*, the more efficient the frontal lobe inhibitory circuits that mediate motor response inhibition; conversely, the greater the *EMG-assessed paratonia*, the less efficient these circuits are.

## 4.2. EMG-assessed paratonia in professional athletes, amateurs and sedentary people

We found that *EMG-assessed paratonia* was lower in professional athletes compared to sedentary individuals, aligning with previous findings from studies using stop-signal and go/no-go paradigms, which demonstrated that professional athletes have superior motor response inhibition compared to their sedentary counterparts [18]. For instance, badminton players exhibited a greater ability to interrupt an already initiated motor response than sedentary controls in stop-signal tasks [29]. Similarly, fencing and taekwondo athletes showed more efficient motor response inhibition than non-athletes, as assessed using a modified version of the stop-signal tasks [30]. Additionally, Nakamoto et al. [31] demonstrated that motor response inhibition was higher in basketball and baseball players than in non-athletes using the go/no-go task.

Previous studies have shown that the ability to inhibit unwanted motor activity is more efficient in professional athletes involved in open-skill sports than in those involved in closed-skill sports. This finding has been attributed to the fact that open-skill athletes must constantly adapt to changing situations influenced by the actions of other players and the environment, requiring a heightened capacity for motor control and inhibition of automatic responses. This constant adaptation is likely to enhance their ability to quickly inhibit automatic or pre-planned responses, allowing for greater flexibility and precision in real-time decision making. In contrast, athletes in closed-skill sports, who train in more stable and predictable conditions, tend to rely less on these inhibitory control processes because their performance relies more on repetition and consistency rather than the ability to adapt quickly to external changes [32, 33]. In line with this, Wang et al. [34] found that professional athletes in open-skill sports, such as tennis, had shorter reaction times in stop-signal tasks (indicating better motor response inhibition) compared to professional athletes in closed-skill sports, such as swimming, who showed similar levels of inhibition to the sedentary population.

In our study, although professional swimmers (closed-skill sports) showed more *EMG-assessed paratonia* (indicating less efficient motor response inhibition) than professional basketball and tennis players (open-skill sports), they still exhibited more efficient inhibition compared to the sedentary population. This discrepancy from the results by Wang et al. [34] could highlight the higher sensitivity of *EMG-assessed paratonia* in distinguishing smaller differences in motor response inhibition between different populations of healthy subjects. Alternatively, it may be because our subjects also participated in open water swimming competitions, where they must cope with the unpredictability of waves, currents, and water temperatures, as well as physical contact with opponents [35].

Furthermore, swimmers demonstrated better motor response inhibition than runners (both closed-skill sports), who, in turn, showed levels comparable to amateurs. *EMG-assessed paratonia* was evaluated in the upper limbs, so this difference could be attributed to the importance of the upper limbs in the technical aspects of each sport. In running, the contribution of the upper limbs is less predominant and mainly limited to maintaining balance and posture, whereas in swimming, the upper limbs are crucial for propulsion [36].

Compared to professional athletes and sedentary individuals, amateurs demonstrated an intermediate level of *EMG-assessed paratonia*, exhibiting higher levels than professional athletes but lower levels than sedentary individuals. These findings suggest that the capacity to enhance inhibitory control is not exclusive to professional sports but is also present in less intense and more occasional activities, such as those typical of amateurs. These results align with previous literature. Studies by Chu et al. [37] and Joyce et al. [38] showed improvements in motor response inhibition, measured with the stop-signal task, after 20 minutes of moderate-intensity running and 30 minutes of low-intensity cycling, respectively. Additionally, a 15-minute low-intensity walking session [39] and a 30-minute moderate-intensity cycling session [40] demonstrated significant improvements compared to the no-exercise control condition using the go/no-go task.

## 4.3. Limitations of the study

A limitation of this study is the lack of a direct comparison between the new method to assess motor response inhibition and traditional experimental protocols for between-group analysis (secondary aim). Only a qualitative and indirect comparison with existing data in the literature was made. To overcome this limitation, future research should implement a more rigorous design, evaluating the same participants using both *EMG-assessed paratonia* and the stop-signal and the go\no-go paradigms for a direct and more detailed comparison of results.

## 5. Conclusion

*EMG-assessed paratonia* can be used in healthy subjects to investigate motor response inhibition, which is one of the most important phenomena among the executive functions of the frontal lobe.

Compared to the methods used to date to study motor response inhibition, the new approach requires less participation from the subject, who is simply asked to remain relaxed during the assessment of muscle tone, rather than being involved in demanding cognitive tasks. This feature could simplify the study of motor response inhibition, allow the phenomenon to be studied in motor contexts that cannot be assessed by the other methods but are relevant in a functional context (e.g. slow and fast, continuous and discontinuous movements), and reduce the risk of learning in longitudinal assessments.

## Supporting information

**S1 Dataset. Puce EMG dataset.**
(XLSX)

## Author Contributions

**Conceptualization:** Lucio Marinelli.

**Data curation:** Luca Puce, Cristina Schenone, Filippo Cotellessa, Daniele Pucci, Nicola Luigi Bragazzi.

**Formal analysis:** Luca Puce, Cristina Schenone, Antonella Tatarelli, Daniele Pucci, Nicola Luigi Bragazzi.

**Investigation:** Luca Puce, Filippo Cotellessa.

**Methodology:** Laura Mori, Carlo Trompetto.

**Project administration:** Carlo Trompetto.

**Software:** Antonella Tatarelli, Daniele Pucci.

**Supervision:** Lucio Marinelli, Antonio Currà, Carlo Trompetto.

**Validation:** Lucio Marinelli.

**Writing – original draft:** Luca Puce, Carlo Trompetto.

**Writing – review & editing:** Lucio Marinelli, Antonio Currà, Laura Mori, Antonella Tatarelli, Carlo Trompetto.

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
