## [Decision Letter · Decision Letter 0]

6 Nov 2024

PONE-D-24-42071EMG-Assessed Paratonia: A Novel Approach to Investigating Motor Response Inhibition in Healthy SubjectsPLOS ONE

Dear Dr. Marinelli,

Thank you for submitting your manuscript to PLOS ONE. After careful consideration, we feel that it has merit but does not fully meet PLOS ONE’s publication criteria as it currently stands. Therefore, we invite you to submit a revised version of the manuscript that addresses the points raised during the review process.

We look forward to receiving your revised manuscript.

Kind regards,

Goran Kuvačić, PhD

Academic Editor

PLOS ONE

“This research was funded by by the project “RAISE - Robotics and AI for Socio-economic Empowerment” and "Fit4MedRob - Fit for Medical Robotics".

Reviewers' comments:

Reviewer's Responses to Questions

**Comments to the Author**

1. Is the manuscript technically sound, and do the data support the conclusions?

Reviewer #1: Yes

Reviewer #2: Yes

2. Has the statistical analysis been performed appropriately and rigorously? 

Reviewer #1: Yes

Reviewer #2: Yes

3. Have the authors made all data underlying the findings in their manuscript fully available?

Reviewer #1: Yes

Reviewer #2: Yes

4. Is the manuscript presented in an intelligible fashion and written in standard English?

Reviewer #1: Yes

Reviewer #2: Yes

5. Review Comments to the Author

Reviewer #1: The article introduces an innovative approach to assessing motor response inhibition through EMG-assessed paratonia in healthy individuals. This approach represents a valuable contribution to the existing body of literature on motor inhibition, expanding beyond traditional methods such as the Go/No-Go and Stop Signal Tasks. The study’s results reveal a significant relationship between physical activity levels and motor inhibition, enhancing its relevance for both neuroscience research and sports science. While the manuscript is well-structured and the methodology appears robust, there are a few areas that would benefit from further clarification and refinement.

1. Introduction:

• The introduction mentions that physical activity has an impact on motor inhibition, but this connection could be explained in more detail. Adding a brief explanation of how physical activity affects the brain circuits involved in motor inhibition (e.g. through training executive functions or improving motor control) could provide a stronger context for the study's hypothesis.

2. Methodology:

• It would be helpful to provide more detail on how self-reported physical activity levels (METs) were verified. Were there any objective measures (e.g., wearable devices) to cross-check these self-reports?

• While the methodology is robust, additional clarity on the rationale for the chosen movement speeds (40 BPM and 100 BPM) would strengthen the manuscript. Why were these specific speeds selected, and how do they relate to real-world movements in the context of sports or daily activities?

• What is the scientific rationale behind the decision to interpolate the data to 101 points for each flexion and extension, instead of using a more conventional number such as 100?

• It might be useful to provide more details on the procedure used to perform the maximum voluntary contraction, which was employed for the normalization of the data.

• It would be helpful to provide a reference for the methodology used to process the raw EMG signal, specifically the band-pass filtering in the 20-450 Hz range, followed by rectification and smoothing using a 4th order Butterworth low-pass filter with a 5 Hz cut-off frequency.

• Similar to before, it would be useful to include a reference to the methodology used to calculate the area under the EMG signal curve using the trapezoidal method, particularly for calculating the area during flexion and extension periods.

• Statistical analysis: The statistical analysis reads fine, but further clarification is recommended. How the sampling was conducted is unclear. Can the authors specify how they recruited the participants (randomly, via snowballing, purposively sampling, convenience sampling, etc.)? Do they consider the participants’ fndings generalizable to the entire population? Can the authors reflect and comment on these aspects?

3. Results:

The presentation of the results is solid and well structured. Furthermore, the presentation of the results through table and figures is well realized, allowing a clear visualization of the main findings.

4. Discussion:

• The difference between ‘open-skill’ and ‘closed-skill’ sports (e.g. swimming and tennis) is discussed, but a clear explanation of why athletes of ‘open-skill’ sports should have more efficient circuits is lacking? This could be related to the neuroplasticity induced by specific physical training, but the explanation is too superficial.

5. Minor Issues:

• There are a few minor grammatical errors throughout the manuscript (e.g., missing articles). A careful proofreading would address these.

Overall, the manuscript is of high quality, and the study provides meaningful contributions to the understanding of motor response inhibition. After addressing the minor revisions, I believe the manuscript will be suitable for publication.

Reviewer #2: The aim of the present manuscript was to investigate whether EMG-assessed paratonia could be used to test motor

inhibition also in cognitively unimpaired individuals, other than in patients with Alzheimer Disease as previously demonstrated. Secondly, it was aimed to test whether the investigation of EMG-assessed paratonia in healthy subjects is

able to reveal a well-known feature of motor response inhibition, specifically its correlation with the

level of physical activity.

The paper is well written; the study methodology is correct and the results are adequately reoprted, demostrating the efficacy of this neurophysiological approach not only in patients with dementia but also in healthy subjects without clinical manifestation of paratonia.

I have no comment

6. PLOS authors have the option to publish the peer review history of their article (what does this mean?). If published, this will include your full peer review and any attached files.

Reviewer #1: No

Reviewer #2: **Yes: **Domenico Antonio Restivo

---

## [Decision Letter · Decision Letter 1]

25 Nov 2024

EMG-Assessed Paratonia: A Novel Approach to Investigating Motor Response Inhibition in Healthy Subjects

PONE-D-24-42071R1

Dear Dr. Marinelli,

We’re pleased to inform you that your manuscript has been judged scientifically suitable for publication and will be formally accepted for publication once it meets all outstanding technical requirements.

Kind regards,

Goran Kuvačić, PhD

Academic Editor

PLOS ONE

Additional Editor Comments (optional):

Reviewers' comments:

Reviewer's Responses to Questions

**Comments to the Author**

1. If the authors have adequately addressed your comments raised in a previous round of review and you feel that this manuscript is now acceptable for publication, you may indicate that here to bypass the “Comments to the Author” section, enter your conflict of interest statement in the “Confidential to Editor” section, and submit your "Accept" recommendation.

Reviewer #1: All comments have been addressed

2. Is the manuscript technically sound, and do the data support the conclusions?

Reviewer #1: Yes

3. Has the statistical analysis been performed appropriately and rigorously? 

Reviewer #1: Yes

4. Have the authors made all data underlying the findings in their manuscript fully available?

Reviewer #1: Yes

5. Is the manuscript presented in an intelligible fashion and written in standard English?

Reviewer #1: Yes

6. Review Comments to the Author

Reviewer #1: in this revised version of the manuscript, the authors have addressed all the raised issues.

I have no other comment.

7. PLOS authors have the option to publish the peer review history of their article (what does this mean?). If published, this will include your full peer review and any attached files.

Reviewer #1: No

---

## [Editor Report · Acceptance letter]

3 Dec 2024

PONE-D-24-42071R1 

PLOS ONE

Dear Dr. Marinelli, 

I'm pleased to inform you that your manuscript has been deemed suitable for publication in PLOS ONE. Congratulations! Your manuscript is now being handed over to our production team.

Kind regards, 

on behalf of

Dr. Goran Kuvačić 

Academic Editor

PLOS ONE